# The Influence of Concentration and Temperature on the Membrane Resistance of Ion Exchange Membranes and the Levelised Cost of Hydrogen from Reverse Electrodialysis with Ammonium Bicarbonate

**DOI:** 10.3390/membranes11020135

**Published:** 2021-02-16

**Authors:** Yash Dharmendra Raka, Robert Bock, Håvard Karoliussen, Øivind Wilhelmsen, Odne Stokke Burheim

**Affiliations:** 1Department of Energy and Process Engineering, Norwegian University of Science and Technology, Kolbjørn Hejes vei 1B, NO-7491 Trondheim, Norway; yash.raka@ntnu.no (Y.D.R.); robert.bock@ntnu.no (R.B.); havard.karoliussen@ntnu.no (H.K.); 2SINTEF Energy Research, NO-7465 Trondheim, Norway; oivind.wilhelmsen@sintef.no

**Keywords:** reverse electrodialysis (RED), anion-exchange membrane, cation-exchange membrane, membrane resistance, membrane resistivity, ion-exchange membrane, hydrogen production, ammonium bicarbonate, low-grade waste heat to hydrogen

## Abstract

The ohmic resistances of the anion and cation ion-exchange membranes (IEMs) that constitute a reverse electrodialysis system (RED) are of crucial importance for its performance. In this work, we study the influence of concentration (0.1 M, 0.5 M, 1 M and 2 M) of ammonium bicarbonate solutions on the ohmic resistances of ten commercial IEMs. We also studied the ohmic resistance at elevated temperature 313 K. Measurements have been performed with a direct two-electrode electrochemical impedance spectroscopy (EIS) method. As the ohmic resistance of the IEMs depends linearly on the membrane thickness, we measured the impedance for three different layered thicknesses, and the results were normalised. To gauge the role of the membrane resistances in the use of RED for production of hydrogen by use of waste heat, we used a thermodynamic and an economic model to study the impact of the ohmic resistance of the IEMs on hydrogen production rate, waste heat required, thermochemical conversion efficiency and the levelised cost of hydrogen. The highest performance was achieved with a stack made of FAS30 and CSO Type IEMs, producing hydrogen at 8.48× 10${}^{-7}$ kg m${}_{mem}^{-2}$s${}^{-1}$ with a waste heat requirement of 344 kWh kg${}^{-1}$ hydrogen. This yielded an operating efficiency of 9.7% and a levelised cost of 7.80 € kg${}_{H_{2}}^{-1}$.

## 1. Introduction

In the imminent transition needed in our energy economy, shifting from fossils to sustainable, a demand for vast amounts of chemically storable energy for transportation and the chemical process industry is emerging. Our energy economy is transcending from burning chemical energy to generate electricity into an energy economy where electricity is used for producing chemicals and chemical energy. In this emerging energy economy, most of the electrical energy is generated from solar and wind power, likely to an extent where utilising waste heat for electricity production will become much less interesting than what has been suggested so far. Utilising industry waste heat for hydrogen production, however, is far more future-oriented. One of very few techniques that can be employed to achieve this is regenerative reverse electrodialysis electrolysis [1].

In a reverse electrodialysis (RED) system, the concentration gradient of salt solutions across an ion-exchange membrane (IEM) acts as a driving force for ions to diffuse through the membrane to create an ionic flux. This flux can be converted into either electrical current or a gas with an appropriate combination of electrolyte rinse solutions and electrodes. Typically, seawater and river water salt solutions are used in RED systems today to produce electricity. Due to geographical constraints and biofouling, closed-loop systems have received increasing attention [2,3,4,5,6,7]. In these systems, the temperatures and pressures of the outlet solutions are modified to restore the initial concentrations [2,8]. Some of these systems use heat to evaporate either the solvent or the solute. The choice of salts in these processes depends on the solubility, the resistivity and the amount of heat required [2]. Ammonium bicarbonate-based reverse electrodialysis (AmB RED) is one such system that has shown potential in developing feasible closed-loop systems [3,9,10]. An example of such an AmB RED system that uses waste heat to generate hydrogen is depicted in Figure 1. This concept will be studied in further detail in this work.

As in many electrochemical energy systems, low resistance and high selectivity are key requirements for the membranes used. For an optimal RED system, the membranes must be thin ($\delta_{m}<$ 100 μm), usually without reinforcement (moderate mechanical properties) and with a low electrical resistance (R${}_{m}$ < 1 Ω cm${}^{2}$) [11]. Determination of the membrane resistance under different working conditions provides insight into loss of efficiency and thus the performance of RED [12]. In this work, we experimentally evaluate the membrane resistance of ten different IEMs. A more thorough introduction and review of methods for measuring the ohmic resistance will be presented in Section 2. The information about the resistance of the IEMs will be used to gauge the influence of the membrane resistance on hydrogen production and waste heat required using a concept similar to the one depicted in Figure 1. Eventually, we will estimate the influence of membrane resistance on the thermodynamic efficiency and levelised cost of hydrogen (LCH).

## 2. Theory and Background

The performance of a RED system is determined by the maximum power produced per unit membrane area and by its maximum efficiency. The process efficiency of a RED system is a major hurdle, and an increase in membrane resistivity decreases process efficiency. Therefore, to optimise the system, the membrane resistivity is a key parameter [13,14]. The ionic resistance of commercially available ion-exchange membranes is often reported as an area resistance, because this quantity is useful for predicting and comparing the performance of the membranes in many applications [15]. Decreasing the thickness of a polymer film tends to reduce the area resistance [11,15]. The thickness of the membrane is essentially fixed by the manufacturing process. This makes evaluation and comparison of the intrinsic ion transport properties of polymer films prepared at different film thicknesses difficult [15]. For homogeneous polymer films, such as the membranes considered in this work, the area resistance can be normalised by film thickness to obtain the intrinsic ionic resistance (i.e., resistivity) of the polymer.

The electrical resistivity of an IEM is a function of the concentration of mobile ions in the IEM and of the mobility of the ions in the membrane phase. The membrane phase ion mobility relates to the ion exchange capacity (IEC), water content (V${}_{w}$) and cross-linkage of the membrane. Typically, an IEM with high IEC, high V${}_{w}$ and low cross-linkage has a low electrical resistivity. This relation is due to the dependence of the electrical resistivity of the ion exchange membrane on the concentration of counterions in the membrane and on Donnan-adsorbed ions. The present work focuses on the use of IEMs for RED working within the typical operating ohmic regime. As this regime is at a relatively low current density, the imposed voltage and current density have a linear relation. Thus, ion depletion in the concentration polarisation boundary layer is not dominant, and a limiting current is not reached. Experimentally, the resistivity of a membrane ${\bar{\rho }}_{mem}$, is found by measuring the amount of current when a certain potential difference is applied over two electrodes.
(1)$${\bar{\rho }}_{mem}=\frac{ASR}{\delta_{mem}}$$
where *ASR* is the area-specific resistance [Ω-m${}^{2}$], $\delta_{mem}$ is the membrane thickness [μm] and the resistivity is an intensive property. There are various methods to evaluate the membrane resistivity. A flow cell configuration is one such method in which a membrane separates two or more compartments of salt solutions with the same concentration as shown as in Figure 2 left [16]. The cell resistance due to the membrane and due to the ionic solutions between the membranes is measured and defined as the stagnant diffusion layer (SDL) resistance [12]. An illustration of the diffusion boundary layers that can be found in the vicinity of the membrane is provided in Figure 2 right.

Another method is a cell configuration in which the membrane is sandwiched between two electrodes with no flow of solution. Here, the ohmic resistance of the electrochemical cell is measured with and without a membrane. The difference between those two values gives the membrane resistance [16]. The resistance measurements can be performed with two, three or four electrodes. In case of two electrodes, one acts as counterelectrode and reference electrode, while the other as working electrode and reference electrode. A three-electrode system includes the working electrode, the counterelectrode and a reference electrode. The potential difference is measured between the reference electrode and the working electrode. In the four-electrode system, there is an additional reference electrode with respect to the three electrode setup. The reference electrodes are connected to a high impedance device, so that in principle there is no current flow through these electrodes. Furthermore, no measurement is needed for the current-generating electrodes [12].

Electrochemical impedance spectroscopy (EIS) gives the frequency-specific impedance of a material. The impedance, Z, is a complex number, defined to be the ratio of the complex potential, E, and the complex current, I. In an EIS measurement, an alternating voltage or alternating current of known frequency and defined amplitude is applied to an electrochemical system. The corresponding response in terms of either current or voltage from the system is measured. When the response to a sinusoidal signal is a sinusoid, the system is said to be a linear system at the same frequency. In case of a complex-valued impedance, the imaginary value is not zero and there will be a phase shift (ϕ).
(2)$$Z=\frac{E(t)}{I(t)}$$
where *E*(*t*) and *I*(*t*) are voltage [V] and current [A] as a function of time *t*. The varying voltage and current with time are defined as
(3)$$E\left(t\right)=E_{0}sin\omega t=E_{0}e^{j\omega t}$$
(4)$$I\left(t\right)=I_{0}sin\left(\omega t+\phi \right)=I_{0}e^{j(\omega t+\phi )}$$
where *E*${}_{0}$ is the voltage in phase, *I*${}_{0}$ is the alternating current in phase and *j* is the imaginary number (*j* = $\sqrt{-1}$). The ϕ is defined as the tangent of the angle between the real and the imaginary impedance. ω is the circular frequency of alternating current and is given by
(5)ω=2πf
where *f* is the frequency.

By using Euler’s formula, the impedance can be defined as
(6)$$Z\left(\omega \right)=\frac{E_{0}e^{j\omega t}}{I_{0}e^{j(\omega t+\phi )}}$$
where the real part and imaginary parts are
(7)$$Re\left(Z\right)\equiv Z^{\prime }=\left|Z\right|cos\phi$$
(8)$$Img\left(Z\right)\equiv Z^{\prime \prime }=\left|Z\right|sin\phi$$

If the phase between current and voltage is zero, this implies that there is no capacitive or inductive response of the system and that there is no imaginary part in the impedance. Macdonald et al. were the first to obtain an exact expression for the small signal impedance for the case of no space charge layers in the absence of an applied potential difference [17]. This resulted in exact equivalent electrical circuits including geometric capacity and frequency dependant impedance. Many researchers opt to a graphical representation of a modelled physical and a chemical process analogous to a circuit electrical diagrams [14], as depicted in Figure 3.

In general, there are three basic elements in such graphical representations: the electrochemical resistor, and the electrical and chemical capacitors. By changing the frequency of the applied AC potential, the resistance of the solution and the electrical double layer (EDL) can be differentiated [12,18,19,20]. While at low applied AC frequency to the membrane system, the resulting electrical equivalent circuit indicates the effect of the diffusion boundary layer (DBL) and the EDLs, at high frequency, the AC reveals the resistance attributed to the membrane polymer itself. That is, the impedance of the IEM is a real value. The impedance data are interpreted in terms of electrical equivalent circuits (EEC) using simple fitting [14]. When a current pulse is applied to an interface, one part is consumed by the EDL charging and the other is used for an interface electrochemical reaction. Randles and Ershler proposed an EEC for a metal electrode impedance [17]. It is composed of two parallel branches with an EDL capacitance in the first one and a reaction resistance in the second. This means that the total electric current can be separated into faradaic and charging currents. Similar principles are used to describe the impedance of an ion exchange membrane. In this work, we use EIS with a direct difference method and a two electrode setup, primarily because it is simple and robust. In addition, it also avoids electrochemical reactions that may occur during the measurement and is more accurate in differentiating the pure membrane resistance from the DBL and the EDL [12,18,20].

### Effect of Solution Concentration, IEM Thickness and Operating Temperature on Membrane Resistance

The membrane resistance depends on the counterion concentration and mobility. IEMs operating at feed concentrations <0.3 M NaCl have displayed significantly increased membrane resistivity. The use of highly concentrated solutions gave a decrease in membrane resistance for an anion-exchange membrane at increasing concentration; the opposite trend was found for a cation-exchange membrane. The difference in the trend was attributed to the density of fixed charge groups and ion exchange capacity, as well as membrane thickness [11,21]. Galama et al. proposed that there are two phases in the membrane: Phase I corresponds to permeation regions with attached fixed charges, where the ionic concentration is determined by electroneutrality and Donnan equilibrium. Phase II corresponds to permeation regions without fixed charges, where the ionic concentration is equal to that in the external bulk solution [12]. The membrane resistance, R${}_{I}$ (Ω cm${}^{2}$) corresponding to phase I, dominates for concentrations higher than 0.3 M. This is the result of the interaction limiting ion mobility in phase I. The concentration-dependent resistance of phase II, R${}_{II}$ (Ω cm${}^{2}$) dominated for the salt concentration lower than 0.3 M. This behaviour is attributed to a change in the ion concentration in phase II. Kamcev et al. proposed that thin bulk solution layers at the membrane surface contribute to the measured resistance [16]. The membrane resistance is sensitive to the salt identity, for which it depends on the counterion identity. Furthermore, the membrane resistance typically increases with increasing hydration free energy of the counterion in the bulk solution, indicating that steric effects are important determinants of membrane ionic resistance. The membrane resistance has displayed a strong inverse correlation with solution concentration below 0.1 M, and remains approximately constant at higher concentrations. The membrane ionic conductivity increases with increase in salt concentration, which was attributed to increased ion concentrations due to weaker Donnan exclusion. The membrane ionic conductivity for CMX decreases due to osmotic deswelling causing decreased ion diffusion coefficients. The apparent resistance is the actual ohmic resistance from the IEMs introduced in a RED stack in operation and is significantly higher than the value from the standard measurement (with 0.5 M NaCl solution) [22]. This effect needs to be clarified and modelled for an improved representation of the RED stack. Several models have been proposed based on experimental data. Veerman fitted experimental data to an exponential function of the form [23]
(9)$$R_{m}=A+B\cdot e^{-rC}$$
where *A*, *B* and *r* are all fitting parameters, and *C* is the solution concentration. Later, Kim et al. pointed out that the membrane resistance is a linear function of the reciprocal of the solution concentration [24]:(10)$$R_{m}\propto \frac{1}{C}$$

Guler et al. found that the membrane resistance does not extrapolate to zero when the membrane thickness becomes zero. Galama et.al. proposed the modified relationship as [22]
(11)$$R_{m}=A+\frac{r}{C}$$

It was found that a route for the fabrication of homogeneous membranes without reinforcement and with reduced thickness yields IEMs with low resistances. Operating at high temperature generally increases the feed resistivity, facilitates ionic mobilities and thus reduces the Ohmic resistance.

## 3. Materials and Methods

The list of membranes used in the present study is presented in Table 1. The membranes are chosen based on their properties: low membrane resistance and high permselectivity in NaCl solution, or lack of literature data on membrane resistance values for ammonium bicarbonate solution for some membranes. The properties listed are extracted from the manufacturers data sheet and from literature.

### 3.1. Membrane Resistivity Measurements

#### 3.1.1. Membrane Equilibration

The experimental setup for direct measurement of membrane resistivities is depicted in Figure 4. We shall next describe the details of how the experiments were carried out. First, membranes were cut in a circular shape with a diameter of 2 cm. Each of these membranes were soaked in a bottle with approximately 200 mL of ammonium bicarbonate solution (Merck, Germany, EMPROVE, 99–101%) in an equilibration concentration (0.1 M, 0.5 M, 1 M, 2 M) for at least 48 hrs without refreshing the solution. The membranes were kept at a temperature of 295 ± 2 K and 313 ± 1 K for room and elevated temperature measurements, respectively. The elevated temperature of 313 K was chosen to reflect the RED system’s improved performance, the expected maximum operating temperature range of the membranes and the temperature-dependent concentration change of the ammonium bicarbonate solution. The counterions listed in Table 1 were exchanged with ammonium ions for cation-exchange membranes and bicarbonate ions for anion-exchange membranes.

#### 3.1.2. Electrode Preparation

The membrane resistance was measured using platinum disc electrodes. The disc electrodes were polarised for uniform surfaces using cyclic voltammetry using the reversible hydrogen electrodes (RHE) prepared in the lab. RHE were prepared in a glass tube with a 0.5 mm diameter platinum wire. An air-tight glass flask of 1200 mL filled with 200 mL of 99 % concentrated H${}_{2}$SO${}_{4}$ diluted in 1000 mL of DI water was used to prepare RHE and disc electrodes using chronoamperometry and cyclic voltammetry. A glass tube was filled with the same solution and then a two-electrode setup was used to produce hydrogen with chronoamperometry using a Gamry 5000 E interface. The experimental settings for this procedure are provided in Table 2.

The chronoamperometry was performed until the tube was 50% filled with H${}_{2}$ gas. The disc electrodes used to measure membrane resistance were made of platinum with an active area of 3.14 cm${}^{2}$ and a thickness of 1 mm. These electrodes were polarised using cyclic voltammetry (CV) under the following conditions the initial and final potential was 0 V. The scan limit, rate and step size were 1.6 V, 20 mV s${}^{-1}$ and 0.5 mV, respectively. The experiment was performed for 60 cycles at maximum current limit of 20 mA.

The CVs were performed with a three-electrode setup with the RHE (explained above) as the reference. Ten to fifteen cycles were performed to have a stable voltammogram which corresponds to a uniform surface. Insulated platinum/copper wires were used to connect the platinum disc electrodes to the potentiostat. The disc electrodes were enclosed in a ceramic hollow cylinder and kept in a thick rectangular metal box. A screw was used to clamp the ceramic cylinder and metal box. A clamping torque of 2 Nm was set for all experiments.

Electrochemical impedance spectroscopy was used to measure the ohmic resistance of the membranes. A set of three measurements were performed for a stack of 1, 3 and 5 membrane layers. The settings for the measurements are as follows: the initial frequency: −10${}^{6}$ Hz; final frequency: −10 Hz, points/decade-15; AC Voltage: −10 mV; Area: −3.14 cm${}^{2}$; and initial delay of 60 s.

Furthermore, a blank cell measurement was performed after every set. In the case of a high-temperature measurement, the setup was kept in a heating oven. For boosting the temperature of the membrane and disc electrodes, an external heating coil was wound along the length on the hollow ceramic cylinder as shown in Figure 4. One of the electrodes was connected to a thermocouple (type K) to measure the temperature of the disc electrode. The measurements were analysed using an equivalent circuit model. The ohmic resistance of the membrane plus the electrodes was then estimated. These resistances were plotted as functions of the membrane thickness. The intercept was treated as the ohmic resistance of the blank cell.

### 3.2. Thermodynamic Model for the RED System

To assess the significance of the measurement results and the resistance of the IEMs, a thermodynamic model for a thermally driven AmB RED based on a closed loop regenerative system developed by Raka et al. was used [26]. For details on the thermodynamic model, we refer to the work in [26]. The performance was evaluated in terms of hydrogen produced and waste heat required for restoring the concentrations.

The unit cell open circuit potential $\left(E_{u.c}^{\mathrm{oc}}\right)$ is the electromotive force driven by concentration difference across an ion exchange membrane with no losses considered. The open circuit potential of an IEM pair placed between two different concentration solutions can be described using the modified Nernst equation: (12)$$E_{u.c}^{\mathrm{oc}}=(\alpha_{\mathrm{aem}}+\alpha_{\mathrm{cem}})\frac{\mathrm{RT}}{F}\ln\left(\frac{\gamma_{c}m_{c}}{\gamma_{d}m_{d}}\right)$$
where α is the permselectivity of IEMs measured at concentration m${}_{c}$ and m${}_{d}$ at a constant temperature of 298 K for a specific membrane. Here, we assume the same α for both membranes. In the above equation, F is the Faraday constant, T is the room temperature and R is the universal gas constant. The activity coefficient of solutions (γ) is a measure of the deviation from ideal solution. The activity coefficients depend on molal salt concentration. There is an internal loss in the RED cell due to its components and operating parameters. This internal resistance consists of ohmic and non-ohmic resistances. The ohmic resistance for a unit cell [Ω m${}^{2}$] is the cumulative sum of membrane and channel (concentrate and dilute) resistances.
(13)$$R_{u.c}=R_{\mathrm{ch},d}+R_{\mathrm{ch},c}+R_{m,\mathrm{aem}}+R_{m,\mathrm{cem}}$$

The channel ohmic resistance (R${}_{\mathrm{ch}}$) is the resistance (Ω m${}^{2}$) due to the conductivity of the feed solution in the channel and spacer geometry. It depends on concentration and is calculated using the molar conductivity of the salt. The actual unit cell potential $\left(E_{u.c}^{\mathrm{act}}\right)$ is the potential across the RED unit cell. The potential drops due to ohmic resistances in the RED unit cell:(14)$$E_{u.c}^{\mathrm{act}}{=E}_{u.c}^{\mathrm{oc}}-R_{u.c}j_{u.c}^{\mathrm{pp}}$$
where the peak power current density ($j_{u.c}^{\mathrm{pp}}$) [A m${}_{mem}^{-2}$] is calculated using the following equation based on Ohm’s law:(15)$$j_{u.c}^{\mathrm{pp}}=\frac{E_{u.c}^{\mathrm{oc}}}{2R_{u.c}}$$

#### 3.2.1. Hydrogen Production

The theoretical amount of hydrogen produced per unit time in the compartment with electrode–electrolyte rinse solution is the hydrogen production rate (${\dot{n}}_{H_{2}}$) [moles m${}_{mem}^{-2}$h${}^{-1}$], which can be calculated from
(16)$${\dot{n}}_{H_{2}}=\frac{j_{u.c}^{\mathrm{pp}}3600}{\mathrm{zF}}\eta_{F}.$$

Here, z = 2 is the ion valence per mole of hydrogen gas, $\eta_{F}$ is the faradaic efficiency. The $\eta_{F}$ considers hydrogen gas losses and signifies that the current density generated is not fully converted to produce hydrogen gas due to various system related losses such as undesired reactions or loss in the form of heat [27]. In a RED system, the loss in faradaic efficiency is due to back diffusion of ions, transport of co-ions and osmosis (i.e., closely related to membrane permselectivity), and ionic short-circuiting in the feed and drain channels. This loss in faradaic efficiency can be as high as 50% in a typical closed loop RED system in comparison with alkaline water electrolysers where the faradaic efficiency ranges from 5 to 25% due to other reasons. Assuming a nearly ideal membrane, the faradaic efficiency is assumed to be 0.90. From Equations (Equation 15) and (Equation 16), we see that the resistance of the membrane is inversely proportional to hydrogen production rate at peak power.

#### 3.2.2. Waste Heat/Regeneration System

The regeneration system restores the outlet concentration of the concentrate and dilute solutions to the corresponding inlet concentrations. It includes a stripping and an absorption process. The air-stripping column decomposes the outlet solution from the dilute compartment to a mixture of ammonia and carbon dioxide gas at 333 K. The absorption unit dissolves the decomposed gases at 293 K at the outlet of the concentrate channel. The heat required for restoring the concentrations to their original concentration is the regeneration heat (q${}_{\mathrm{reg}}$). The total amount of thermal power required to strip a volume flow rate of Q${}_{\mathrm{lc}}$ [m${}^{3}$ s${}^{-1}$] per unit membrane area of ammonium bicarbonate salt from the dilute solution channel is computed from
(17)$$Q_{\mathrm{reg}}=q_{\mathrm{AmB}}Q_{\mathrm{lc}}\;A_{\mathrm{mem}}^{\mathrm{tot}}\times 3600,$$
where $q_{\mathrm{AmB}}$ is the specific thermal duty [kWh m${}^{-3}$] required to decompose ammonium bicarbonate solution into its components NH${}_{4,(g)}$ and CO${}_{2,(g)}$. The value was estimated using the relation from Bevacqua et al. with inlet and outlet concentration C${}_{1}$ and C${}_{2}$ from the stripping column, respectively:(18)$$q_{\mathrm{AmB}}=a_{1}e^{a_{2}\cdot C_{1}}-a_{3}C_{2}^{a_{4}}+a_{5}C_{1}^{a_{6}}C_{2}^{a_{7}}$$

Here, *a*${}_{1}$ to *a*${}_{7}$ are fitting parameters that are functions of C${}_{1}$ [10].

#### 3.2.3. Levelised Cost of Hydrogen

The model used to estimate the levelised cost of hydrogen (LCH) for the proposed system was developed previously by Raka et al. [26]. The parameters that have been changed for the present study are stated below, and we refer to Raka et al for further details [26]. Parameters used for this study are as follows: C${}_{mem}$ is cost of membrane, 5–150 € m${}^{-2}$; C${}_{heat}$ is cost of waste heat, 0.01 € kWh${}^{-1}$; $\eta_{ele}$ is electrolysis efficiency, 0.9; and t${}_{mem}$ is membrane life, 10 years.

## 4. Results and Discussion

The membrane area resistances were estimated using equivalent circuit modelling of the data from the impedance measurement using a simple equivalent circuit as shown in Figure 3. The resistance in series (R${}_{m+ele}$) is associated with the ohmic resistance from the membrane and the electrode. A blank cell measurement was performed to estimate the electrode resistance. This value was subtracted from R${}_{m+ele}$ to estimate the membrane resistance values. In the following section, the measured values for R${}_{mem}$ will be reported and discussed.

### 4.1. Influence of Thickness on Membrane Resistance and Membrane Resistance at Elevated Temperature

The membrane resistance is not only a material property, it is a ratio between the membrane thickness and its conductivity. With increasing membrane thickness, the length of transport pathway increases, and the membrane resistance increases proportionally. For all of the IEMs considered in this study, we observe a linear relationship between the membrane resistance and the membrane thickness. This has been shown for anion-exchange membranes (AEMs) at 295 ± 2 K and 313 ± 1 K in Figure 5 and Figure 6, and for the cation-exchange membranes (CEMs) at the same temperatures in Figure 7 and Figure 8. The membrane resistance is here found to increase with increasing thickness. A linear regression was used to fit the resistance as a function of thickness, and the coefficients of these polynomials and their uncertainties are provided in the Appendix A in Table A1, Table A2, Table A3 and Table A4. Though the linear equation describes the effect of thickness on the resistance reasonably well, we observe that there is a high double standard deviation in the resistivity (the slope of the equation). This is likely to be associated with the high variation in the IEM thickness, which is difficult to measure for swollen membranes, especially for the APS type (AEM). The increase in resistance due to increase in thickness is because an ion has to traverse further through the tortuous path inside of the membrane. Even though the AEMs are thinner compared to CEMs, in general, the CEM have lower resistances compared to the AEM. This may be due to the mobility of NH${}_{4}^{+}$ ions, which is very high compared to the mobility of HCO${}_{3}^{-}$ ions, as shown in Table 3.

The Fumasep membranes with reinforcement—FASPET and FKSPET—have higher resistances compared to FAS and FKE. The increased resistance can be explained from the increased thickness (polyester reinforcement). FAS and FASPET follow a similar trend when it comes to the resistance as a function of the thickness. From Figure 5, Figure 6, Figure 7 and Figure 8, it is difficult to evaluate the membranes based on their thickness normalised membrane conductivity or membrane ohmic resistance, as they give a different prioritisation of which membranes should be preferred. Although a thinner membrane may show lower conductivity at different concentrations, the measured membrane resistance of the thinner membranes can be lower than the thicker membranes with highest conductivity. This is because the conductive membrane is thicker.

In general, the membrane resistance decreases with increasing temperature. This can be explained on the basis of the increase in ionic mobility through the membrane which increases with temperature.

### 4.2. Influence of Concentration on Membrane Conductivity

Figure 9 shows how the conductivities of the different AEMs and CEMs change with concentration. The first conclusion that can be made on the basis of these results is that the conductivities of the CEMs are typically significantly higher than those of the AEMs. The mobility ratio of ammonium ions to bicarbonate ions is 1.67, this may explain higher conductivities in CEM membrane compared to AEM.

The electrolyte concentration influences the various IEMs differently. In the case of the CEMs, CMF and CSO exhibit the highest conductivities of all the membranes at both temperatures examined. For the AEMs, APS and DSV have the highest conductivities at both temperatures. APS and CSO display a clear trend, where the conductivity increases with increasing concentration.

AMV, DSV and FKE have the lowest conductivities at 0.5 M and 1 M, respectively, and the conductivities increase/remain constant with further increase in concentration. This may be due to loss of water through the IEM or shrinkage, causing loss of conduction paths. In case of FAS, FASPET and CMV, the conductivity decreases with increase in concentration above 0.5 M, and it is lowest at 2 M. CMF shows almost constant conductivity with increase in concentration. The exact values of the mean conductivities of the membranes are reported in the Appendix A in Table A5 and Table A6.

### 4.3. Influence of Membrane Resistance on H${}_{2}$ Production rate and Specific Waste Heat Required Q${}_{req}$

After the membrane characteristics have been determined experimentally, it is of interest to study how they influence the performance of a RED stack for generation of hydrogen by use of waste heat. Based on the measured values, we find that a lower ohmic resistance for the membranes gives higher hydrogen production rate and lower waste heat required per kg of hydrogen. The article compares the hydrogen production rate and specific waste heat required for membranes with the highest conductivity (APS and CMF) with the least resistance (FAS and CSO). The highest conductivity was found for a stack made with APS and CMF for 2 M/0.1 M feed solutions concentration. The hydrogen production rate and specific waste heat required based on APS and CMF membranes were found to be 6.51 × 10${}^{-7}$ kg m${}^{-2}$s${}^{-1}$ and 419 kWh kg${}_{H_{2}}^{-1}$, respectively. For a stack made of FAS30 and CSO membranes as AEM and CEM with the least area specific resistance (Ω m${}^{2}$) for 2 M/0.1 M feed solutions concentration at room temperature, the corresponding hydrogen production rate and specific waste heat required is 8.48 × 10${}^{-7}$ kg m${}^{-2}$s${}^{-1}$ and 344 kWh kg${}_{H_{2}}^{-1}$. The hydrogen production from a RED stack with NaCl solution operating at sea water/river water concentration of 0.6 M/0.0015 M was theoretically estimated to be 9.64 × 10${}^{-7}$ kg m${}^{-2}$s${}^{-1}$ and artificial NaCl solution with concentrations 4 M/0.017 M was experimentally found to be 4.94 × 10${}^{-8}$ kg m${}_{ele}^{-2}$s${}^{-1}$[27,28]. Compared with the proposed system, the reasons for low hydrogen production rate can be low permselectivity, high area-specific membrane resistance, high overpotentials and effect of activity coefficients. In comparison, the conventional electrolysis such as alkaline electrolysers operating at atmospheric conditions the hydrogen production rate was found to be 2 × 10${}^{-4}$ kg m${}^{-2}$s${}^{-1}$ [29,30]. Therefore, an increase in the membrane resistance decreases the electrochemical potential of the stack, and thus the resulting peak power current density decreases. This decrease in current density decreases the number of hydrogen moles produced due to electrolysis of water as well. From the previous sections, it is clear that a thinner membrane can have low area-specific resistance but can be relatively less conducive, for example, FKE. On the other hand, a thicker membrane can have more resistance but can be relatively highly conductive, for example, CMF. The results signify that the membrane’s thickness plays a vital role in improving the hydrogen production rate. In the thermodynamic model to calculate the peak power density produced, we consider area-specific resistance, and we emphasise ohmic resistance rather than conductivity. Therefore, a stack made of CMF/APS has a higher area-specific resistance compared to CSO/FAS and was not chosen as optimum. Furthermore, the decrease in the operating current density causes a decrease in the amount of salt transported through a membrane due to electro-migration, and decreases the amount of water dragged along due to electro-osmosis. As the salt flux decreases, the concentration of dilute solution at the outlet decreases. The decrease in the dilute solution concentration decreases the waste heat required per unit volume of dilute solution (q${}_{th}$ [kWh m${}^{-3}$]). However, the total waste heat required per kg of hydrogen produced increases as the volume of the dilute solution at reduced current densities increases due to a increased number of unit cells required to develop the potential in a stack required for electrolysis of water.

### 4.4. Influence of R on η and LCH

An increase in the membrane resistance decreases the thermodynamic efficiency, η, and increases the levelised cost of hydrogen (LCH) as shown in the Figure 10. For a stack made of FAS30 and CSO membranes as AEM and CEM with the least area specific resistance (Ω m${}^{2}$) for 2 M/0.1 M feed solutions concentration at room temperature, the thermochemical conversion efficiency was found to be 9.7% and the LCH was estimated to be 7.80 € kg${}_{H_{2}}^{-1}$. The highest conductivity was found for a stack made with APS and CMF for 2 M/0.1 M feed solutions concentration. The corresponding thermodynamic efficiency and LCH estimated were 8.00% and 9.99 € kg${}_{H_{2}}^{-1}$, respectively. The membranes with a lower resistance give a higher thermochemical conversion efficiency, lower waste heat required per kg of hydrogen and lower levelised cost of hydrogen. Thus, with an increase in membrane resistance, the thermochemical conversion efficiency decreases and the LCH increases significantly.

The increase in membrane resistance decreases the performance of the cell by decreasing operating potential and peak power current density. Furthermore, with an increase in membrane resistance, the amount of heat required to restore the concentrations to original increases due to the increased amount of solution from the stack needed for same amount of hydrogen production capacity. Therefore, a decrease in efficiency can be observed. In the case of LCH, the number of membranes required increases with an increase in the membrane resistance and thus the investment cost and membrane replacement cost increases. Moreover, an increase in the volume of the outlet solutions increases the capacity of the regeneration system, which in turn increases the cost. All these factors cause an increase in LCH.

## 5. Conclusions

In the present work, the membrane resistance/conductivity were estimated for membranes soaked in a ammonium bicarbonate salt solution at various concentrations and different temperatures. It was found that cation exchange membranes (CEMs) have higher conductivities than anion exchange membranes (AEM) for nearly all of the concentrations examined (0.1 M, 0.5 M, 1 M and 2 M). For use in combination with reverse electrodialysis, the highest thickness normalised conductivity was found for a stack consisting of the APS type AEM and the CMF type of CEM.

In a closed loop ammonium bicarbonate reverse electrodialysis system that uses these type of membranes in combination with waste heat to generate hydrogen, the hydrogen production rate and specific waste heat required based were estimated to be 6.51 × 10${}^{-7}$ kg m${}^{-2}$s${}^{-1}$ and 413 kWh kg${}_{H_{2}}^{-1}$. This resulted in a thermodynamic efficiency of 8.00% and an estimated levelised cost of hydrogen of 9.99 €kg${}_{H_{2}}^{-1}$. Of the membranes examined in this work, the FAS30 type of AEM and the CSO type of CEM were found to have the smallest area specific resistance. With a closed loop reverse electrodialysis system with these membranes, the hydrogen production rate and specific waste heat required are estimated to be 8.48 × 10${}^{-7}$ kg m${}^{-2}$s${}^{-1}$ and 344 kWh kg${}_{H_{2}}^{-1}$, respectively. This resulted in a thermodynamic efficiency of 9.7% and an estimated levelised cost of the hydrogen of 7.80 € kg${}_{H_{2}}^{-1}$.

## Figures and Tables

**Figure 1 membranes-11-00135-f001:**
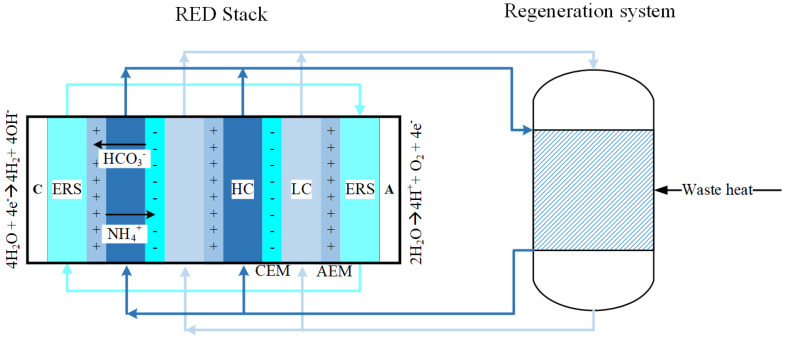
Schematic of an ammonium bicarbonate reverse electrodialysis cell with a thermally regenerative system. Here, C is the cathode where the hydrogen evolution reaction (HER) occurs, A is the anode where water is produced. Both of these electrodes are in contact with electrode rinse solution made of 1 M sodium bicarbonate. The “+” marked symbol denotes anion exchange membrane and “−” marked symbol denotes cation exchange membrane. The concentrate and dilute feed solutions are described as HC and LC, respectively.

**Figure 2 membranes-11-00135-f002:**
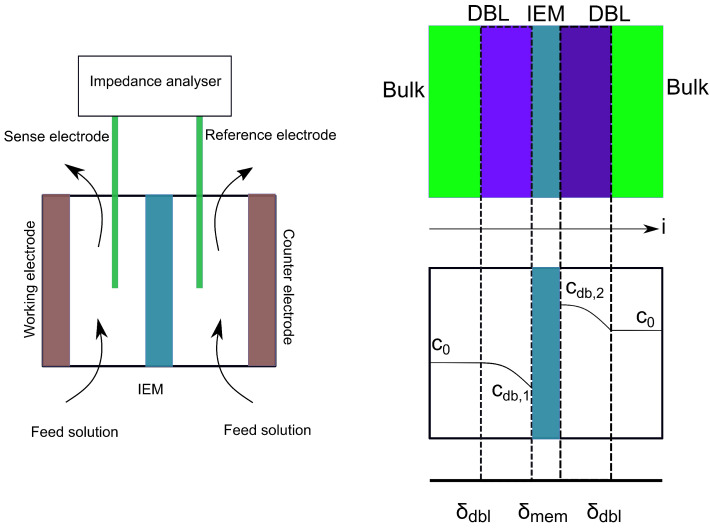
A flow cell configuration measuring membrane resistance using electrochemical impedance spectroscopy (**left**). Diffusion boundary layer (DBL) near the surface of a cation-exchange membrane and the salt concentration distribution in different layers at steady state (**right**). The current direction is to the right.

**Figure 3 membranes-11-00135-f003:**
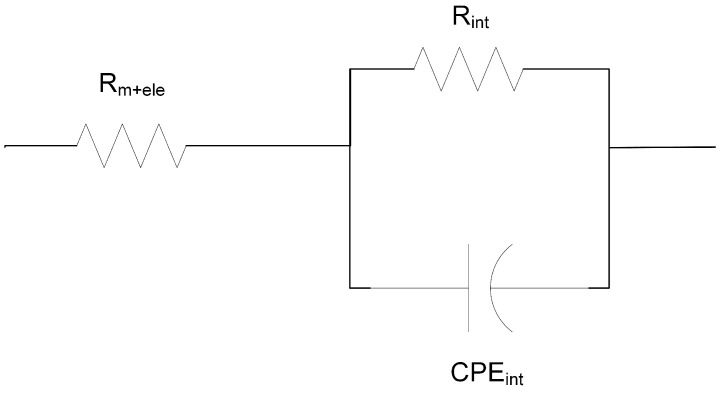
Equivalent circuit used for the fitting of electrochemical impedance spectroscopy (EIS) measurements. The resistance (R${}_{int}$) and constant phase element (C${}_{int}$) represent the interface between electrode and solution layer on the ion-exchange membrane (IEM) (i.e., solution–electrode interface). The ohmic resistance of the IEM and the electrode is represented by R${}_{m+ele}$.

**Figure 4 membranes-11-00135-f004:**
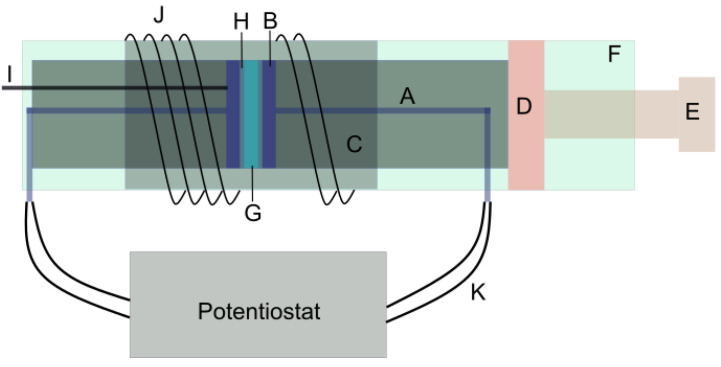
Experimental setup for direct membrane resistance measurement using two electrodes and electrochemical impedance spectroscopy. A. Platinum wire, B. Platinum disc electrode, C. Ceramic casing, D. Stopper, E. Tightening screw, F. Metal casing, G. IEM, H. Solution thin film, I. K-type thermocouple, J. Heating coil, K. Cables.

**Figure 5 membranes-11-00135-f005:**
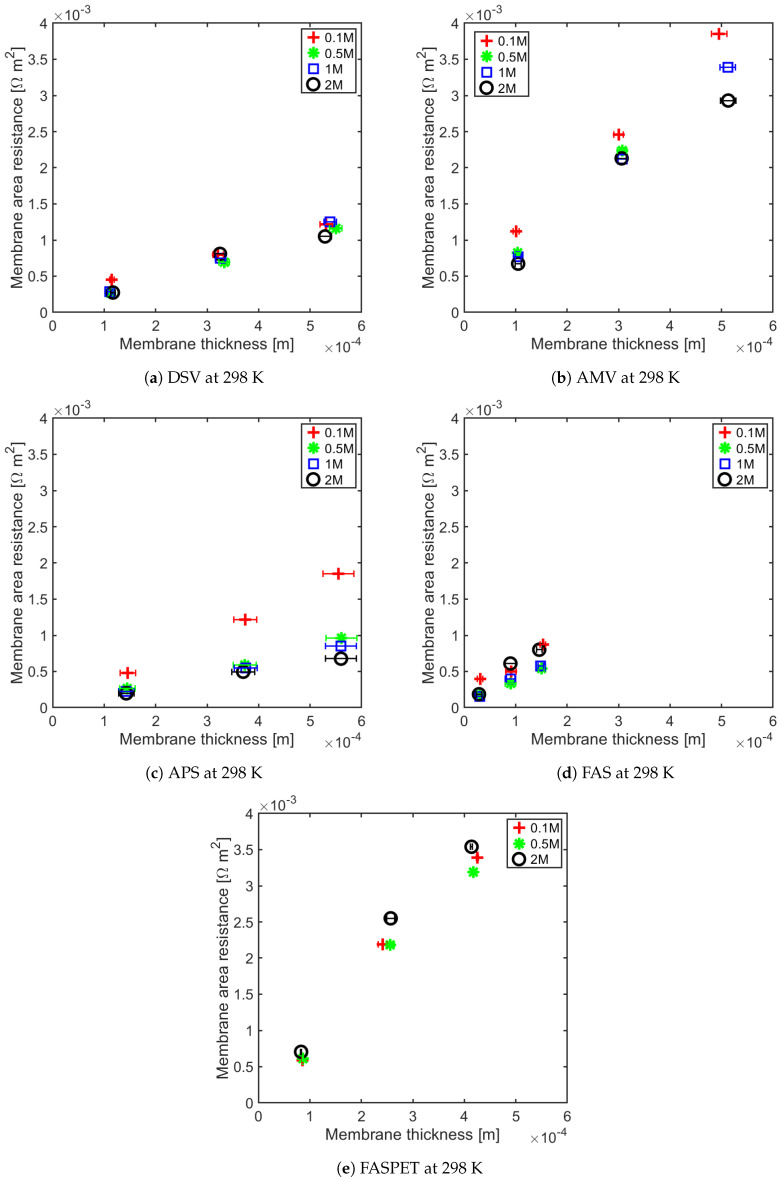
Anion exchange membrane (AEM) resistance as a function of thickness for different concentrations at 298 K.

**Figure 6 membranes-11-00135-f006:**
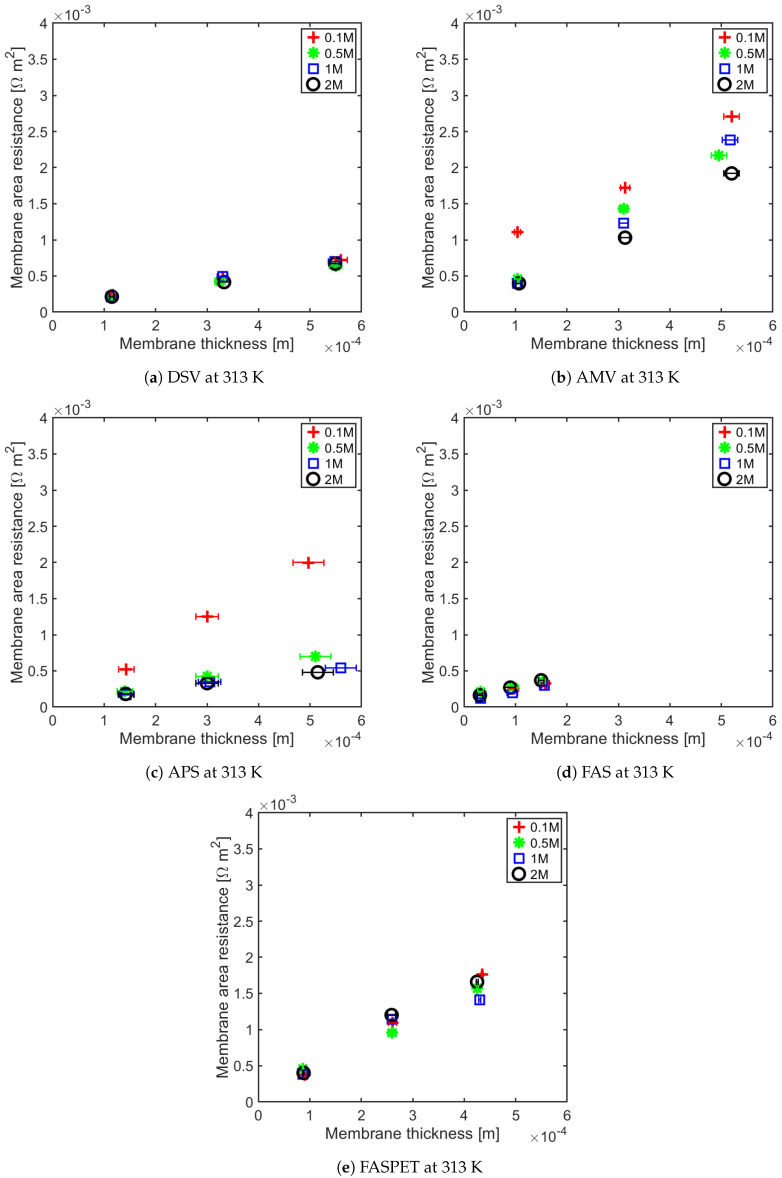
AEM resistance as a function of thickness for different concentrations at 313 K.

**Figure 7 membranes-11-00135-f007:**
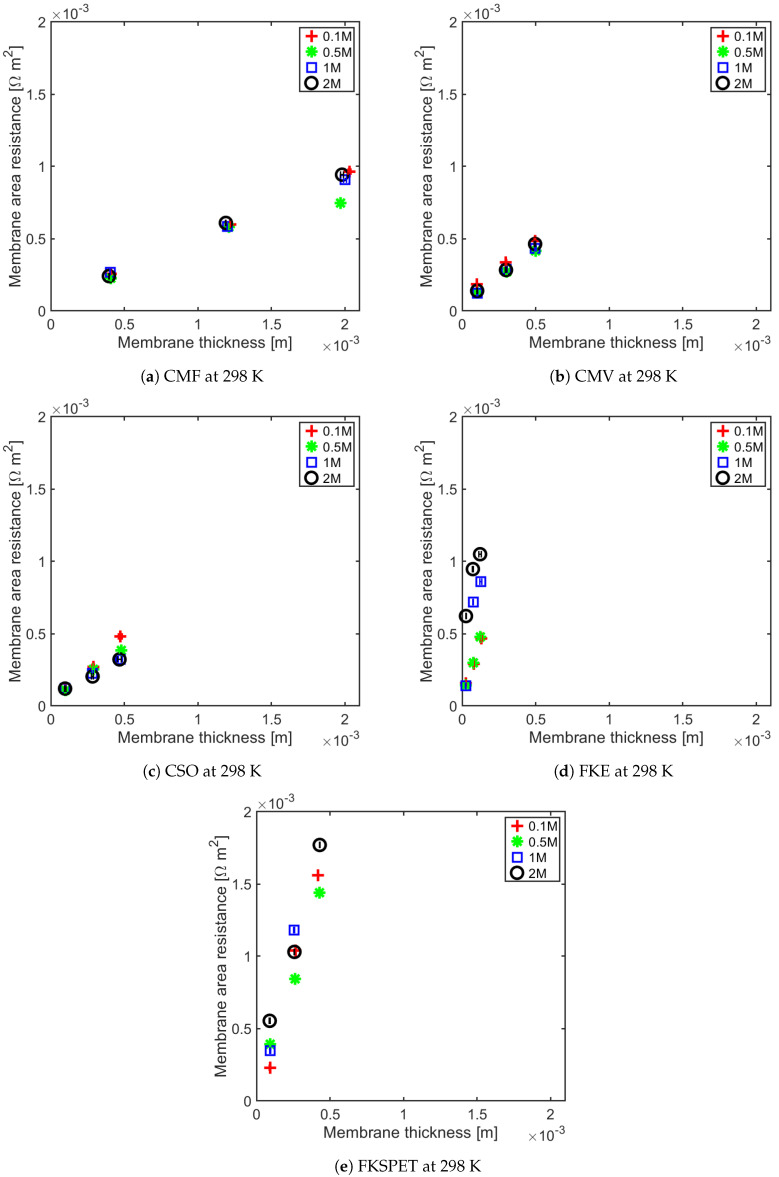
Cation exchange membrane (CEM) resistance as a function of thickness for different concentrations at 298 K.

**Figure 8 membranes-11-00135-f008:**
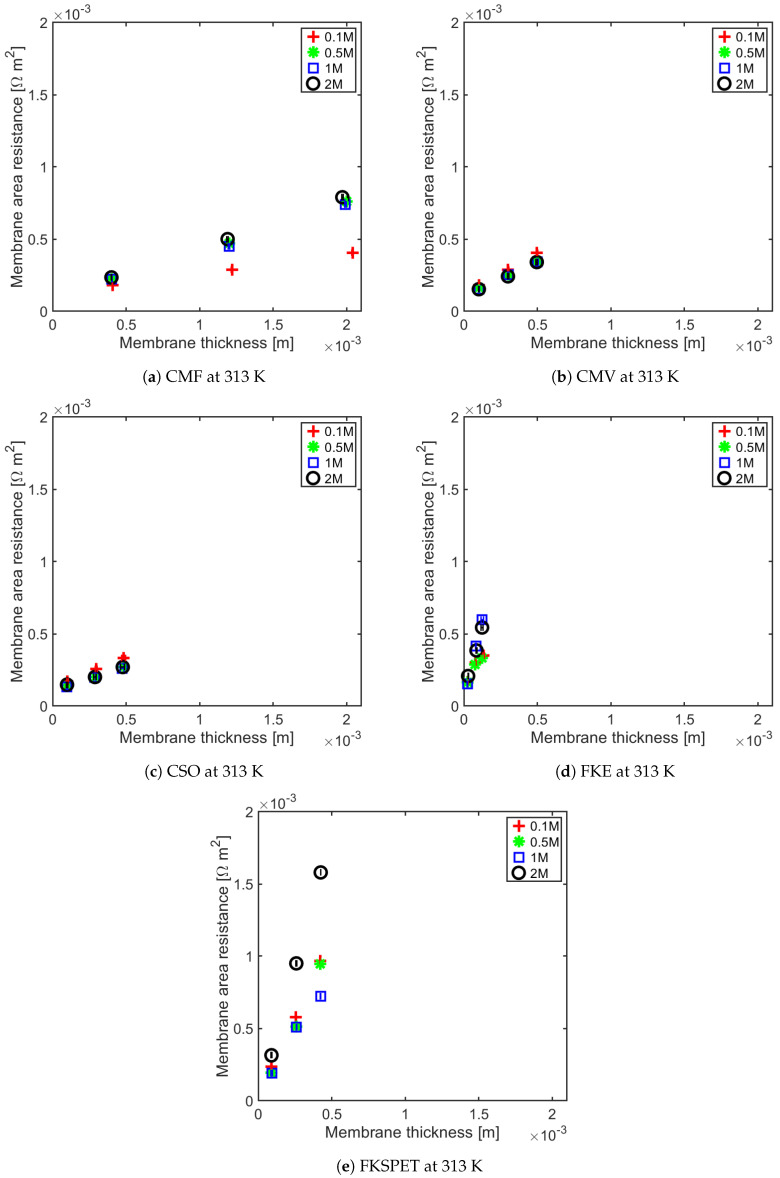
CEM resistance as a function of thickness for different concentrations at 313 K.

**Figure 9 membranes-11-00135-f009:**
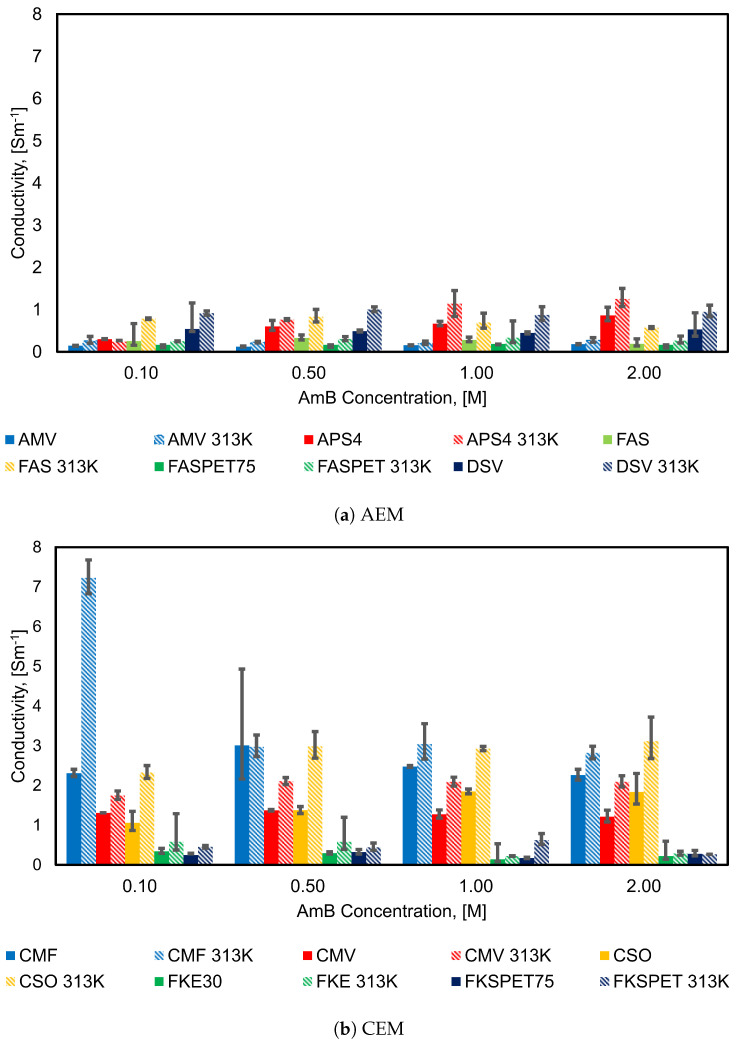
Effect of the solution concentration on the IEM conductivity.

**Figure 10 membranes-11-00135-f010:**
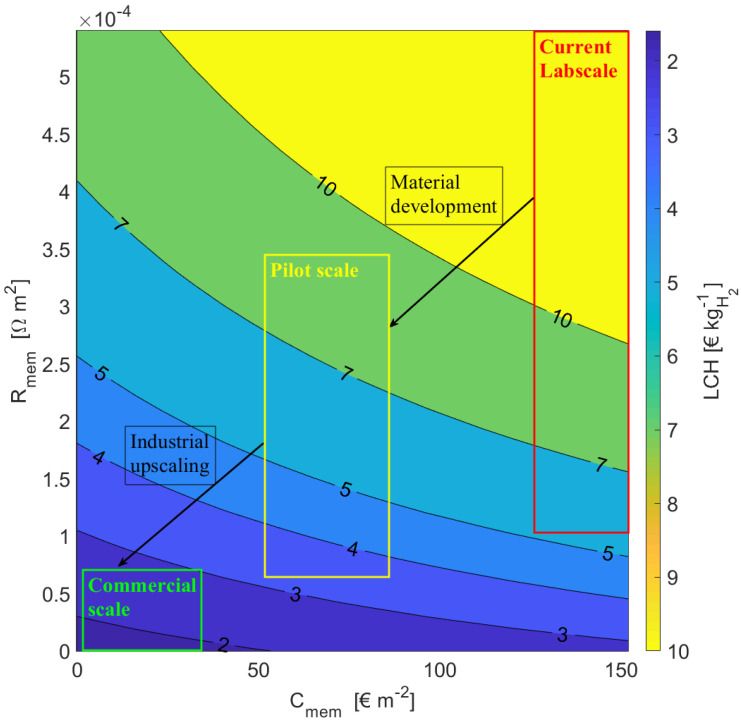
Levelised cost of hydrogen (LCH) as a function of membrane resistance for different membrane cost. Here, we assume the membrane resistances R${}_{aem}$ = R${}_{cem}$. CSO and FAS30 permselectivities were measured and used as input. The membrane cost range assumed for lab, pilot and commercial scale is 130–150, 50–80 and less than 30 € m${}^{-2}$, respectively. The cost of heating is assumed to be 0.005 € kWh${}^{-1}$[26]. We assume for the pilot scale, that the material development will lead to cheaper and less resistive membranes, i.e., SPEEK or carbon-based membranes, and that for the commercial scale these membranes will be produced at large scale (60 × 10${}^{8}$ m${}^{2}$ per year) and reduced thickness (25 μm and maintaining the mechanical strength) which will in turn reduce the cost and resistance further [26,31,32].

**Table 1 membranes-11-00135-t001:** Overview of the membranes examined in this work. The membrane resistance and permselectivity values extracted from manufacturer’s data sheet are tested in 0.5 M NaCl at 298 K and 0.5 M/0.1 M NaCl at 298 K, respectively.

IEM	Type	Thickness μm	Fixed Charge Group	Material	Counter-ion	Permselectivity	Resistance $\times 10^{-4}\Omega$ m${}^{2}$	IEC meq g${}^{-1}$	SD (wt) meq g${}^{-1}$	Ref
FKE	CEM	28–33	-SO${}_{3}^{-}$	-	H${}^{+}$	0.965–0.986	1.6–2.46	1.35–1.36	12–27	[21]
FKSPET	CEM	74–87	-SO${}_{3}^{-}$	-	H${}^{+}$	>0.95	2.5	1–1.25	-	*
FAS	AEM	27–33	-	-	Br${}^{-}$	0.894–0.9	1.03–2	1.1–1.85	8–19	[14,21]
FASPET	AEM	72–85	-	-	Br${}^{-}$	>0.9	<3	1–1.5	-	*
DSV	AEM	95–121	-		Cl${}^{-}$	0.899	2.3	1.89	28	[21]
AMV	AEM	110–150	-N(CH${}_{3}$)${}_{3}^{+}$	PS/DVB/CMS	Cl${}^{-}$	0.873–0.96	2.8–3.15	1.78–1.9	17–19.8	[14,21,25]
CMV	CEM	101–150	-	PS/DVB	Na${}^{+}$	0.91–0.988	1.03–1.1	2–2.4	20–30	[14,21,25]
CSO	CEM	100	-	PS/DVB	Na${}^{+}$	0.923–0.97	2.29–3	1.04	16	[25]
CMF	CEM	440	-	-	H${}^{+}$	> 0.95	2.5	-	-	*
APS	AEM	138–150	-N(CH${}_{3}$)${}_{3}^{+}$	PS/DVB/CMS	SO${}_{4}^{2-}$	0.884	0.68–0.7	0.29	147	[21,25]

* Manufacturer’s data sheet.

**Table 2 membranes-11-00135-t002:** Parameters for the chronoamperometry procedure.

Parameter	Value	Unit
Electrode Area	3.14	cm${}^{2}$
Pre-step Voltage	0 vs. $E_{ref}$	V
Pre-step Delay Time	0.5	s
Step 1 Voltage	−2.5	V
Step 1 Time	200	s
Step 2 Voltage	0.1	V
Step 2 Time	5	s
Max Current	200	mA
Limit I	200	mA cm${}^{-2}$
Equil. Time	5	s

**Table 3 membranes-11-00135-t003:** Properties of the ions considered in this study.

Ion	NH${}_{4}^{+}$	HCO${}_{3}^{-}$
Hydrated radius [nm]	0.331	0.439
Charge density [mC cm${}^{-3}$]	1.05	0.45
Average polarisability [a.u]	7.91	23.7
Ionic mobility [cm${}^{2}$V${}^{-1}$s${}^{-1}$]	7.71 × 10${}^{-4}$	4.59 × 10${}^{-4}$

## Data Availability

The data presented in this study are available in Appendix A.

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
