# Peer review of "The Influence of Concentration and Temperature on the Membrane Resistance of Ion Exchange Membranes and the Levelised Cost of Hydrogen from Reverse Electrodialysis with Ammonium Bicarbonate"

_membranes, 2021, doi:10.3390/membranes11020135_

Round 1

Reviewer 1 Report

This paper compared the various ion-exchange membranes by the effect of the solution concentration and temperature for hydrogen production from RED.

The author insisted that the lower ohmic resistance of the ion-exchange membrane enhanced the hydrogen production rate from RED. And authors investigated various types of ion-exchange membrane to define the optimum condition under different solution concentrations and temperatures.  Besides, they compared thermochemical conversion efficiency and cost estimation in optimum ion-membranes.

The reviewer partly agrees with the author's opinions. However, the reviewer thinks that the obtained results are not sufficient to support the author’s insists.  Also, some research approaches seemed to error. However, the reviewer thinks that this manuscript showed a good approach, but the description and sufficient information is not enough. As well as, I should authors considered a problem as below. For these considerations, I think this manuscript should be rejected.

Critical concerns

  1. This manuscript is article of research not Reviews paper, it has a lot of non-essential information in section 2. Author need to delete section 2, or integrate the section 1 and 2. The author need to modify the following manuscript sections by instruction for authors
  2. How the author selected the operation temperature? 295 and 313K is not enough to determine the temperature effect, the reviewer thinks to need at least three over of temperature factor.
  3. The reviewer thinks that the parameter information (table 1,2,4, and 5) is able to change into the text instead of the table. Whereas, the result of the concentration and temperature seemed to need to make a table to the comparison between the different factors, because it is the main result of this manuscript, and the axis of graphs is a different scale.
  4. The author compared with various Fumasep membranes in this study. Why did you choose these membranes? it was used in previous research? In reviewer think that the membrane should be selected commonly used in RED reactor in previous papers.
  5. Author mentioned that a lower ohmic resistance for the membranes give higher hydrogen production rate -. However, the author chooses the optimum CEM as CMF. Why the CMF is better than CMV? In figure 6,8 the CMV obtained lower membrane area resistance and thickness by different concentrations than CMF.  Whereas, CMV showed relatively lower conductivity than CMF in Figure 9. The conductivity is more important than the ohmic resistance? if yes, Author needs to describe the correlation between conductivity and ohmic resistance.

Minor concerns

  1. Author should be match the figure (i.e., figure/fig) between caption and text
  2. Author need to use one type of unit (i.e., kg/m or kg m-1)
  3. Why Author used italic at sub-section? Using one of the following fonts: Times, Arial, Courier, Helvetica, Ubuntu, or Calibri.
  4. Author need to check APSV or ASP4 in Figure 9 and text. These words used mixed.

Reviewer 2 Report

Dear authors,

I was pleased to read your paper, already made up as paper.

About the form the paper was offered:

Normally manuscripts are offered as WORD document making it easy for reviewing purposes. I have also communicated this to the Editor.

However, because of your excellent writing style I managed to review your paper in the way it was offered.

This makes I can directly turn to the content. 

In your work I propose to compare the energy needed for H2 generation via PEM electrolyzer. This needs about 43kWh/kg H2 (see e.g.: Hydrogen production by PEM electrolysis - A review by S. Shiva Kumar, V. Himabindu, Materials Science for Energy Technologies 2 (2019) p. 442-454.

Then let us examine Figure 1 and the half reactions at they should read:

At the anode we have 2H2O = 4H+ + O2 + 4e-        (1)

At the cathode we have 4H2O + 4e- = 2H2 + 4OH-   (2)

The half reactions as given in Figure 1 do not correspond to the half reactions for water electrolysis. 

The redox solution (1 M NaHCO3) has a pH of around 8.3 (this is for 0.1 M NaHCO3). This means the cathode reaction will not be the reduction of protons.

The anode reactions as given now is the oxygen reduction but that is not an anodic reaction. Therefore please use reactions (1) and (2) as given.

The type of membranes must be given in Figure 1. This was not done. Please mention AEM and CEM. Figure 1 is also too small right now. 

Then we focus at Table 3: 

Clearly you have made some obvious mistakes in Table 3:

> CEM membranes have -SO3- groups (sulfonate groups) and not sulfate groups.

> The perm selectivity of DSV is 0.899 and not 89.9.

> The resistance values are given in Ohm.cm2 and not in Ohm.m2.

On page 8 first alinea: The ohmic resistance of the membrane plus the electrodes was the estimated. 

Figure 10: How could we come to a commercial scale? Please compare the 344 kWh/kg H2 with commercial PEM electrolyzer that needs about 43 kWh/k H2

Since all proposed corrections are minor issues, I consider it a minor revision.

I look forward to your revised version,

Kind regards,

Reviewer

Reviewer 3 Report

One half part, the authors experimentally evaluated the membrane resistance of ten different ion exchange membranes under different electrolyte concentrations and different temperatures. The other half part, the authors investigated the influence of membrane resistance on H2 production rate. This paper provides some basic data for the commercial membranes under different test conditions. But there are several ambiguities that should be clarified prior to publication on Membranes.

  1. There are so many commercial ion exchange membranes, please specify the reasons for the chosen of these ten kinds of membranes. For instance, the CSO is a monovalent ion exchange membrane, not a general type of mono-polar ion exchange membrane, why this kind of membrane was used for the RED process?
  2. One main aim of the present study is to investigate the effect of temperature on the membrane resistance. But only two temperatures (298K, 313K) were investigated. To give a general trend of the influence of temperature, more data with different temperature should be provided. Please provide the data of membrane resistance under more different temperatures.
  3. Some are so many tables with little information, please merge several tables or even delete them to save the spaces.
  4. In section 2.2, the title of this section is the “Effect of solution concentration, IEM thickness and operating temperature on membrane resistance”, but the effect of temperature on the membrane properties was not even discussed. Please add the related contents of the effect of temperature on the membrane properties.
  5. In table 3, it has wrong for the unit of resistance of the membrane. For perm-selectivity, there are so many digits for the data. Please also provide the conditions for the determination of permselectivity and the membrane resistance.
  6. In table 5, please explain the means of each symbol.
  7. The authors need to rearrange of the scale of values in the y-axis, such as Figs. 7d, 8d, and 9a. It is not clear and you should narrow the range.
  8. In section 4.3, Influence of membrane resistance on H2 production rate, the authors should compare your H2 production rate with that in the literature. In the current case, it is difficult to evaluate the significance of your hydrogen production rate.
  9. In figure 10, the authors give many assumptions for the membrane cost under lab, pilot and commercial scale. At least, there should be some references to support that assumptions. Otherwise, the proposed scenario for the levelised cost of hydrogen is not convinced at all.

Round 2

Reviewer 1 Report

This manuscript overall improved.
The reviewer thinks the manuscript is qualified for publication for Membranes journal.

Author Response

The reviewer is not satisfied with the authors' response on my second concern.  The authors have the revised the description of temperature effect on membrane resistance as "membrane resistance at elevated temperature". This temperature (303 K) is just a little higher than the room temperature; it still need further specify the reason for the reason for the chosen of this temperature.

We thank the reviewer for pointing out the lack of precision in our formulating again. Following are the reasons for the choice of temperature.

  1. From equation 12 and 13, the increase in operating temperature increases the open-circuit potential and decreases the solution's resistance in the channels. This improves the performance of the systems, i.e., hydrogen production rate.
  2. Considering the operating pH, IEMs chosen in this study can operate at the maximum 313 K, suggested by the manufacturers.
  3. Further, ammonium bicarbonate is a thermolytic solution at any temperature above 313 K, it decomposes into ammonia and carbon dioxide gases, thus changing the concentration of ionic species in the solution. This changes will affect the measured results.

Hence to avoid damaging IEMs and the inaccuracy in concentration we measure the membrane resistance up to 313 K. We have added the following sentence to chapter 3.1.1., lines 181 ff. in the manuscript to reflect this better:

The elevated temperature of 313 K was chosen to reflect the RED system's improved performance, the expected maximum operating temperature range of the membranes and the temperature-dependent concentration change of the ammonium bicarbonate solution.

Reviewer 3 Report

The reviewer is not satisfied with the authors' response on my second concern.  The authors have the revised the the description of temperature effect on membrane resistance as "membrane resistance at elevated temperature". This temperature (303 K) is just a little higher than the room temperature; it still need further specify the reason for the reason for the chosen of this temperature. 

Author Response

(The authors gave the same response as above.)
